# Recruiting, training and managing a sensory panel in odor nuisance testing

**Paweł Turek**  *

Department of Non-Food Product Quality and Safety, Cracow University of Economics, Rakowicka, Cracow, Poland

* turekp@uek.krakow.pl

## Abstract

A growing level of environmental awareness in societies has led to an increased interest in the odor nuisance. Residents' complaints have ultimately revealed the need to develop adequate research methods that would ensure objective measurements, thus shedding more light on this complex problem. One of the possible ways to conduct an objective odor assessment is to use a sensory panel in the tests. This paper presents the procedure for selecting and training the sensory analysis team especially for the purposes of odor nuisance testing. Several useful suggestions for conducting studies that involve a sensory panel are also provided. This in turn can prove a useful tool, supplementing the EN 13725 standard within the scope of building and convening a sensory panel. In this paper a performance comparison of two sensory panels is also discussed: one that received a basic training of 20h, solely based on the guidelines of the EN 13725 standard and another, whose members completed an extended training of 60h, based on the standards and guidelines used in the field of sensory analysis. It has been shown that acquiring more knowledge and developing certain competences in the field of sensory analysis may contribute to an increase in the overall and individual precision of determinations as adequate sensory training improves the panel's performance.

## 1. Introduction

Research on the unpleasant odors from industrial or communal facilities and their impact on the environment has long been established as a subject of particular interest in many countries, including Germany, France, the Netherlands, Great Britain, Japan, the USA and Canada [1]. Nevertheless, due to the multifaceted nature of the problem, to date no uniform EU legislation concerning this issue, in the form of a directive or a set of guidelines, has been introduced. A very complex relationship between the olfactory sensitivity of individual people and the concentration of chemical compounds in the air, their type as well as meteorological and topographic conditions influencing the spread of odors cause unceasing debates in scientific community. These disputes concern mainly the methodology of measuring odor nuisance.

Dynamic olfactometry is currently the most widespread technique used in Europe for quantifying odors emitted by industrial plants, and its methodology has been standardized by the

**Data Availability Statement:** All relevant data are within the manuscript.

**Funding:** This publication was financed with subsidies for maintaining the research potential granted to the Cracow University of Economics.

**Competing interests:** The authors have declared that no competing interests exist.

EN 13725 standard [2, 3]. Yet, the Polish version of the standard: *Air quality. Determination of odor concentration by dynamic olfactometry* [4] was published almost 15 years ago.

However, even this standard has met with various reservations as to the presented research procedure. One of the main objections related to the use of human senses as a measuring apparatus is primarily the matter of repeatability of results and a certain subjectivity of the assessors. It is oftentimes noted that the perception of smells is strictly speaking an individual matter: the same odor can cause different reactions depending, for example, on the assessment of the source of the smell and individual sensitivity of a given person [5]. At the same time, it may seem that the ongoing development of instrumental methods such as gas chromatography, ion mobility spectrometry (IMS) and infrared spectrometry (IR), ensuring the detectability of odorant concentrations at the human detection threshold level, should contribute to the rapid development of appropriate odor nuisance assessment procedures. And yet, the high sensitivity of these methods most often applies to pure gases [6], whereas the actual odor nuisance is a mixture of various compounds. The difficulty in recognizing and determining the degree of intensity of odors by subjects in mixed samples is the root cause of the imperfect correlation between olfactometric determinations and instrumental odor measurements [7].

Due to synergistic effects, in certain combinations, these compounds may increase the magnitude of the sensation in comparison with the expected one, as calculated on the basis of a simple summation of the effects of individual stimuli. This makes the sensory approach particularly useful in overcoming these problems [8].

Therefore, it is worth considering the use of a team of individuals who will act as a measuring sensor, as such procedure can provide valuable source of information on odor nuisance [9, 10] and, at the same time, it can become the basis for various remedial actions to be taken in view of residents' complaints.

By far the most important factor in obtaining precise results of sensory determinations is a properly recruited and trained sensory panel. Such team is able to provide insightful and precise information on the received stimuli. Of course, all panelists should obtain adequate training to be able to exhibit good discriminatory abilities and to reproduce the ratings on identical samples [11]. Such a team is already calibrated at the stage of selection of candidates, and the reliability of the determinations is constantly monitored, similarly to other fields of scientific analytics. Many procedural guidelines are included in international ISO standards [12]. It also appears that sensory determinations can be used to verify complaints from residents of the affected areas as well as in the assessment of newly established industrial plants or of other forms of economic activity that may have a negative impact on the environment in terms of odor nuisance [13]. The aim of this paper is to present the recruitment and training procedure of the members of the sensory panel, which was created especially for the purposes of research on odor nuisance in the city of Kraków. The said sensory panelists participated in several months of odor nuisance studies. The data obtained from the determinations of the sensory panel then became the subject of scientific analyzes that were published in renowned international journals [14–16] or presented at international conferences [17].

This paper does not focus on detailed checks of sensory sensitivity and the principles of selecting candidates as described in the EN 13725 standard (since it provides clear guidelines within the said scope); instead, the emphasis is placed on the requirements included in the scientific standards of sensory analysis [18]. As such, it provides some useful tips on conducting odor nuisance tests with the participation of a sensory panel. The presentation of methods for creating a sensory panel based on the experience and expertise coming from the field of sensory analysis methodology may not only supplement the EN 13725 standard, but also provide support while carrying out determinations regarding odor nuisance with the use of static, dynamic and field olfactometry methods. The choice of procedures used in the selection of

candidates depends largely on the type of sensory panel and the methods with which it is going to work. At the same time, by developing certain competences and gainig knowledge in the field of sensory analysis, an increase in the precision of determinations may be achieved, as sensory training increases the panel's overall performance [19].

That is why, one of the most crucial aspects of this paper is the discussion of team performance comparison results on the basis of the study of two sensory panels: one that completed a basic training of 20h, organized around the guidelines and regulations contained in EN 13725, and another–whose members received extended training of 60h, based on the standards and guidelines used in the field of sensory analysis.

## 2. Ethics statement

Only adults participated in the recruitment to the sensory team. Participation in the tests and assessments was voluntary. Informed, written consent was obtained from each participant in the study. Each of them could withdraw their consent without providing any justification. Each participant also consented to the processing of their personal data in accordance with Article 6 of Regulation (EU) 2016/679 of the European Parliament and of the Council of 27 April 2016 on the protection of natural persons with regard to the processing of personal data and on the free movement of such data, and repealing Directive 95/46/EC (General Data Protection Regulation). All research procedures were in line with the Code of Good Practices in Universities elaborated by the Polish Rectors Foundation and adopted by the Plenary Assembly of the Conference of Rectors of Academic Schools in Poland (CRASP) on 26 April 2007 and with the ethical standards of the Cracow University of Economics adopted by the resolution of its Senate (No. 38/2011). All participants obtained a detailed description of the test and were informed about the possible range of substances that would be assessed. All methods were performed in accordance with the relevant guidelines and regulations included in the International Organization for Standardization (ISO) concerning sensory analysis. Each of the assessors was obliged to report any indispositions and allergies and if such was the case, the subject did not participate in the tests.

## 3. Recruitment

The selection of sensory panelists to perform odorometric assessments is generally carried out on the basis of the information provided in the EN 13725:2003 standard. The general requirements for the recruitment given in EN 13725 are broadly in line with the basic recommendations that can be found in EN ISO 8586 „*Sensory analysis—General guidelines for the selection, training and monitoring of selected assessors and expert sensory assessors.*"

Clause 6.7. of the EN 13725 standard provides a detailed code of conduct for assessors and assessment team members. To qualify for the assessment panel, prospective panelists should adhere to the following code of conduct:

- they should be motivated to perform the task diligently (based on the author's experience in sensory research, it seems that a high level of motivation is one of the key elements, especially during longer periods of teamwork)

- they should be available during the entire assessment session

- they should be committed for a sufficiently long period of time as to make the tracking and control of their sensory assessment history possible

- to avoid any possible disturbances in their own perception or the perception of other panelists, they should limit the use of cosmetics (perfumes, creams, deodorants) with intense odors on the day of the assessment session,

- 30 minutes before and during the sensory assessment, the panelists should not be allowed to smoke, eat, drink (with the exception of water) or use chewing gum

- they should not take part in the assessment if they have a runny nose or any other condition that affects the perception of smell,

What is more, during sensory analysis, the panelists should not communicate with each other about the results of their own assessments (information on the correctness of the performed determinations is provided after the end of the session).

It is quite important already at this stage to determine the number of panelists. It is usually assumed that initially at least two or three times more individuals should be recruited that the expected final number of the members of sensory panel. According to the requirements of the EN 13725, the minimum size of the team should not be lower than four (clause 6.7.3). On the other hand, elsewhere in the document it is mentioned that the participation of five sensory panelists can be considered optimal (clause 8.5). Taking into account the guidelines contained in other standards in the field of sensory analysis, the number of four individuals constituting a sensory team as specified in EN 13725 seems insufficient. Depending on the type of research, the standards suggest recruiting six assessors [20–22], eight assessors (BS EN 1622, 2006), or even ten–[23–25]. It is also worth noting that in sensory analysis, the guidelines for the minimum number of panelists often depend on the method that is to be used in the research. For example, fifteen or more assessors are required for the triangle method (often used in olfactometric measurements). Therefore, correct determination of the minimum size of the sensory panel in odor testing might prove quite difficult. Moreover, an increase in the number of members automatically entails an increase in the cost of research and it also extends the duration of the determinations themselves. Although the tests can be performed by a team of four assessors, thus already meeting the requirements set out in the EN 13725 standard, studies show [26] that by increasing the size of the panel, the repeatability of the tests may be visibly improved; hence, it is worth considering the sensory team of at least six assessors.

## 4. Final selection (qualification)

In the pre-selection phase, the candidates should be assessed from the perspective of their fitting the future sensory panel. Basic information regarding health condition, availability and readiness to perform tests, as well as certain psychological predispositions to conduct sensory assessments should be collected from the candidates. In this initial phase, some basic sensory tests are also conducted, e.g., the assignment of samples to different groups or the arrangement of the samples according to the changing intensity of the stimulus. The sensory panel that is to perform odor tests should be selected from among the candidates with average sensory sensitivity, thus representing the general population of consumers. At this stage, individuals exhibiting very high sensitivity thresholds, which indicate a low sensory sensitivity, should not be considered as potential panelists.

According to the guidelines specified in the EN 13725 standard, in order for the assessor to become a member of the sensory panel, the geometric mean of the individual threshold assessment ($ITE^1_{subst}$) should be within the range from 62 $\mu g/m^3$ to 246 $\mu g/m^3$ (from 0.02 $\mu mol/mol$ to 0.08 $\mu mol/mol$), whereas the antilogarithm from the standard deviation $S_{ITE}$, calculated from the logarithms ($\log_{10}$) of individual threshold assessments expressed in units of mass concentration of n-butanol, should not be lower than 2.3. These requirements, although very detailed, focus on assessing the sensitivity of the assessors to merely one substance—n-butanol. While understanding the need to adopt such an assumption to simplify the recruitment and qualification to the sensory panel, it should be noted that in the light of general requirements

for the members of sensory teams [23], the training procedure should be much more extensive. During such training, it is worth using the substances found in field studies on odor nuisance, starting from the substances with lower odor nuisance (such as ethyl alcohol, nitrogen dioxide, bromine, phenol, hexanol) to substances with significant or high odor nuisance (e.g., cresol, chlorine, sulfur dioxide, xylene, acetaldehyde, skatole, octanol, ammonia, butyric acid, pyridine) [6, 27–29]. During the training sessions, the prospective assessors should become familiar with various sensory determination methods, starting with simple assignment methods, recognition methods and differential methods (paired comparison test, triangle test, duo-tri test). Such stimuli should initially be introduced as single odors. Subsequently, mixed samples, composed of two or more components in different proportions, can be presented to the assessors for testing. Diversification of the training not only increases the panelists' level of abilities (becoming familiar with various methods of sensory analysis, using various scales, being able to adequately describe the impressions), but it also shapes their motivation by raising interest in this particular field of study. Considering that during the sensory analysis training, assessors should not be given more than 10 odors per session [30], between the sessions more theoretical knowledge related to the methodology and the factors influencing sensory determinations could also be shared with the participants. As Baryłko-Piekielna and Matuszewska [12] rightly point out, familiarizing candidates with the basics of the functioning of the senses, as well as the mechanism of sensing sensations and its physiological and psychological conditions is a prerequisite for the proper understanding of the tasks set before the panel and putting sensory determinations into a certain context. An important element enhancing the assessors' motivation is adequate feedback [31] on the results of the testing, as each team member should be aware that their work matters. The assessors should be observed and monitored throughout the training process, especially during determinations. Tracking the performance of the assessors can also become the basis for deciding on the need to provide any additional training as the proper calibration of the assessors is a key element of the repeatability of the results [32].

## 5. Sensory team building—an example from the research on odor nuisance

During the conducted research on odor nuisance, the measurements were based on the determinations made by the sensory panel. One of the many interesting questions was to determine how extended training can affect the performance of the entire panel; that is why, the qualification phase was divided into two stages. The first one covered the time from the recruitment through pre-selection, training and qualification, as based on the EN-13725 standard. A person could be qualified for the sensory panel when the requirements for sensitivity to n-butanol were met. In the second stage, by using more specialized knowledge in the field of sensory analysis, the panelists' skills related to performing tests were gradually expanded.

### 5.1 Stage 1

The number of individuals recruited to the study was 54. Participation in the tests and the assessments was voluntary. Each of the assessors was obliged to report any indispositions and allergies and if such was the case, the subject did not participate in the tests. All participants obtained a detailed description of the test and were informed about the possible range of substances that would be tested. The preparation of the panel to conducting sensory determinations as well as all qualification tests were carried out in the sensory analysis laboratory of the University of Economics in Kraków. All recruited individuals were asked to arrange and rank 5 n-butanol samples of different intensity. The critical value of the Spearman's rank correlation coefficient (which translates into the ability to correctly rank the samples) was set at 0.7.

Eventually, 23 subjects obtained Spearman's rank correlation coefficient $r_s$ greater than or equal to 0.7. These individuals were then invited to an interview during which the plan for further training and the rules for making determinations were presented. The training lasted 20 hours. After several training sessions, 12 assessors (9 women and 3 men aged 23–44 years) met the basic requirements of EN 13725 regarding the individual n-butanol detection threshold and were qualified for the sensory team. At the end of the training, the ability of the prospective panelists to rank 10 samples differing in the intensity of n-butanol odor was checked.

## 5.2 Stage 2

The aim of the second stage (lasting 60 hours) was to improve the ability to assess odor samples and to increase the precision and repeatability of the determinations. No selection of candidates was performed as all sensory panel members met the requirements for n-butanol sensitivity. One of the panelists, due to unforeseen circumstances, did not participate in the extended training, therefore the analysis presents the assessments of 11 people. The training was based on the sensory analysis standards [23, 30, 33–36] and the literature on the subject [18, 37]. During the determinations, the assessors learned about the smells that may occur in the field (field olfactometry) and cause odor nuisance. The following substances were used: trimethylamine (cas-number 75-50-3) dimethylsulfide (75-18-3), butanoic acid (107-92-6), acetic acid (64-19-7), ethyl acetate (141-78-6), styrene (100-42-5), benzaldehyde (100-52-7), cedryl acetate (77-54-3), geosmin (16423-19-1), lsobutylquinolein (65442-31-1) benzothiazole (95-16-9), 2-phenylethanol (98-85-1), cis-3-heksen-1-ol (928-96-1). Commercially available natural (food and industrial) products were also used, so that the panelists could recognize and

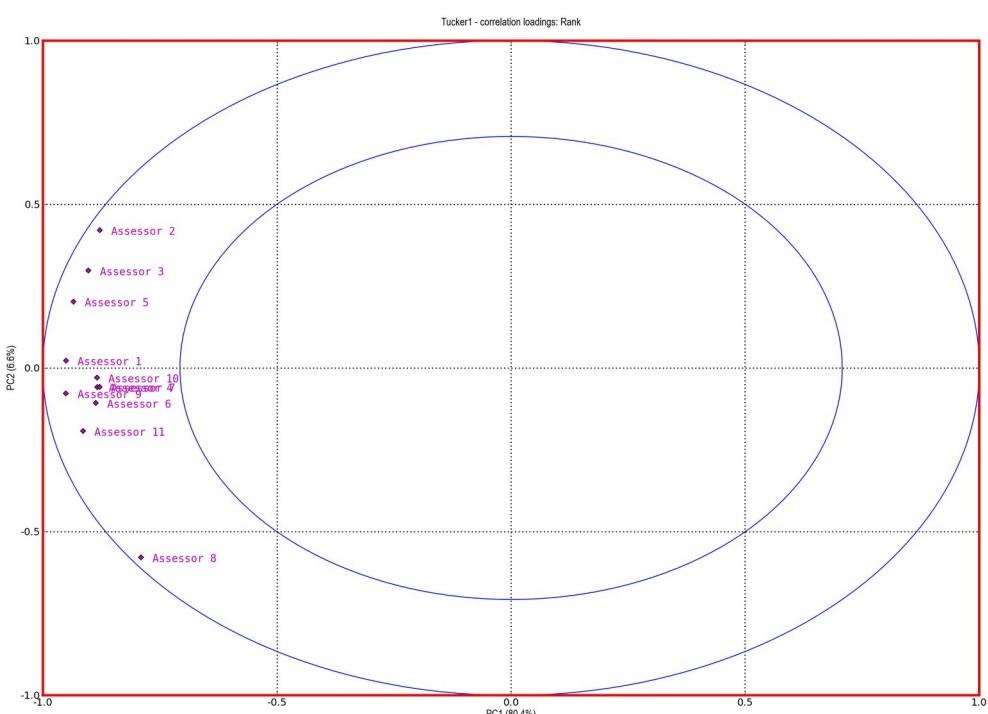

**Fig 1. Tucker-1 correlation loading plots—all assessors stage 1.**

verbally define smells. For this purpose, the assessors were presented with a profile assessment card, which is generally used in field determinations, found in the manual of The Nasal Ranger Field Olfactometer. There are eight groups of odors given in this card (Floral, Fruity, Vegetable, Earthy, Offensive, Fishy, Chemical, Medicinal), which lists a total of 110 odor descriptors. During the training, the Odor Sensitivity Test kit (St. Croix Sensory) was used. The triangle test was also applied to test the ability to detect small differences in the intensity of odor samples. In the final stage of the extended training, a test was carried out for the panelists to identify various odors. The samples used in the test had the same appearance, color and other features, they only differed in smell and code number. The methodology was based on the EN ISO 5496 standard. In each session, 10 different substances were provided for testing. There were three sessions in total. The percentage of correct answers was calculated on the basis of the rating scale given in the standard (ISO 8586, 2012), where individual answers are assigned the following number of points; 3 points—for correct identification or description of the most common associations, 2 points—for describing the sample using general terms, 1 point—for identifying or describing appropriate associations after a previous discussion, 0 points—for a wrong answer or no answer. Each of the assessors could obtain a maximum of 90 points. It is assumed that individuals who do not obtain 65% of points in such tests are not eligible to become *selected assessors* (*selected panelists*). All of the subjects who participated in the tests obtained a result above 72 points, which constituted 80% and exhibited adequate level of the task completion. At the very end, the ability to rank 10 samples differing in the intensity of n-butanol odor was checked again. The samples were presented to the panelists in different sequences.

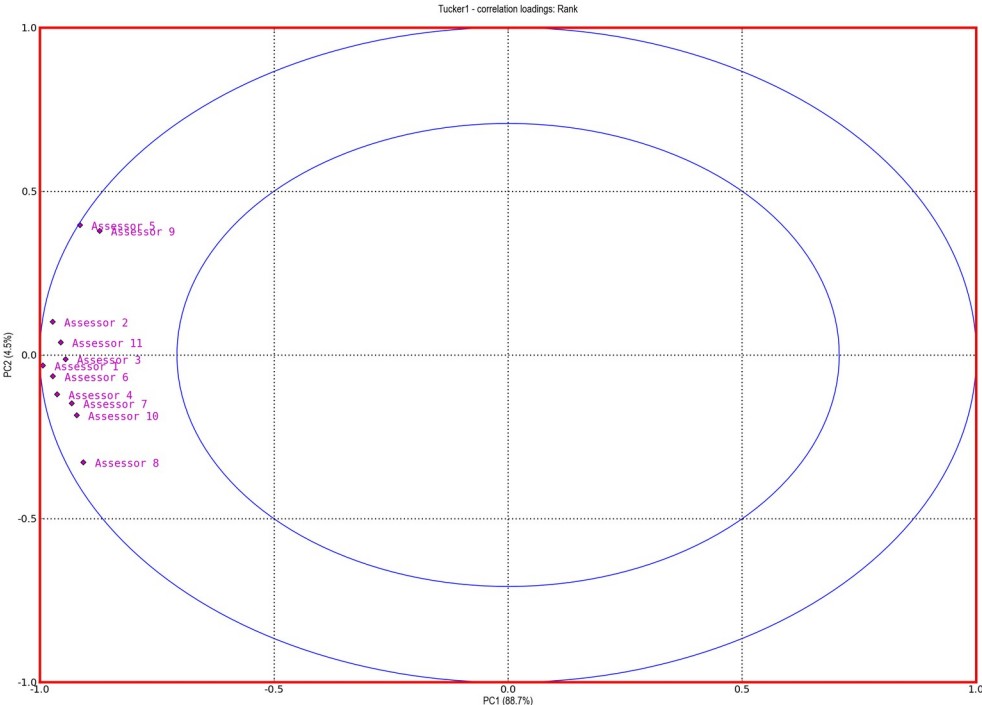

**Fig 2. Tucker-1 correlation loading plots—all assessors stage 2.**

## 6. Sensory panel performance comparison

To evaluate the performance of the panels, PanelCheck free software was used [38]. Figs 1 and 2 (Tucker -1 plots to check agreement within the panel) and Figs 3 and 4 (eggshell plot–to check the sample ranking across assessors) show graphical representations of the results of all team members who received training in stages 1 and 2. The test concerned the ability to rank 10 samples differing in the intensity of n-butanol smell. For a well-trained and calibrated panel the correlation loadings of the sample under investigation should be close to the outer ellipse with all panelists clustered closely together [39].

Bearing in mind that the sensory efficiency is not a constant value and is subject to certain fluctuations, it is worth noting that the results of all assessors, both in the first and the second stage, are similar to the outer ellipse, which indicates the correct calibration of the team. A more detailed analysis shows that the assessments made after additional training are less scattered, which is evidence that panel members are more consistent with each other when it comes to sample ordering. A graphical comparison of the results of all assessors in a single graph is shown on the Eggshell Plot. Such visualization allows for easy identification of those panelists whose results differ significantly from the others. The eggshell plot visualizes

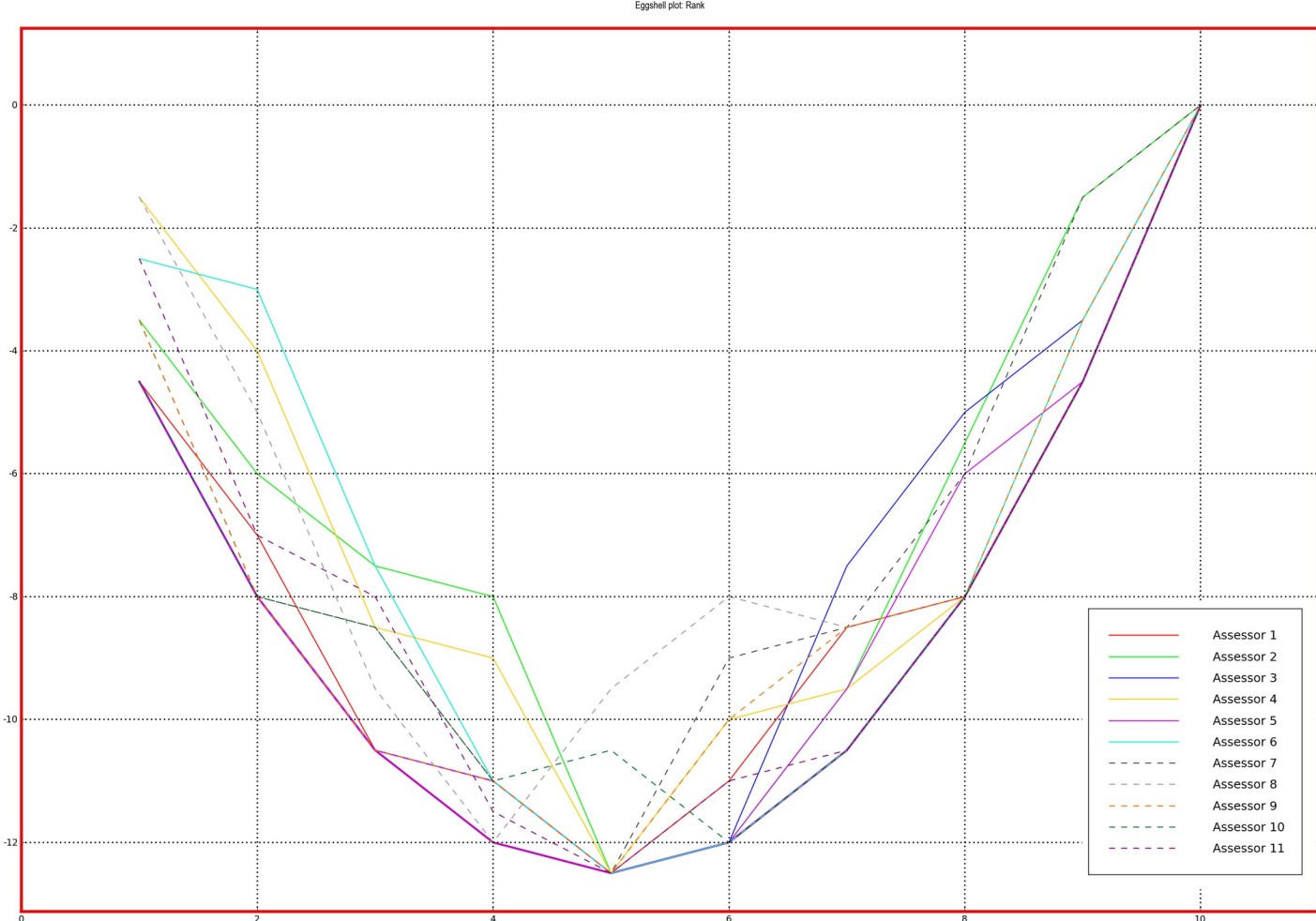

**Fig 3. Eggshell plot for each of the assessors–stage 1.**

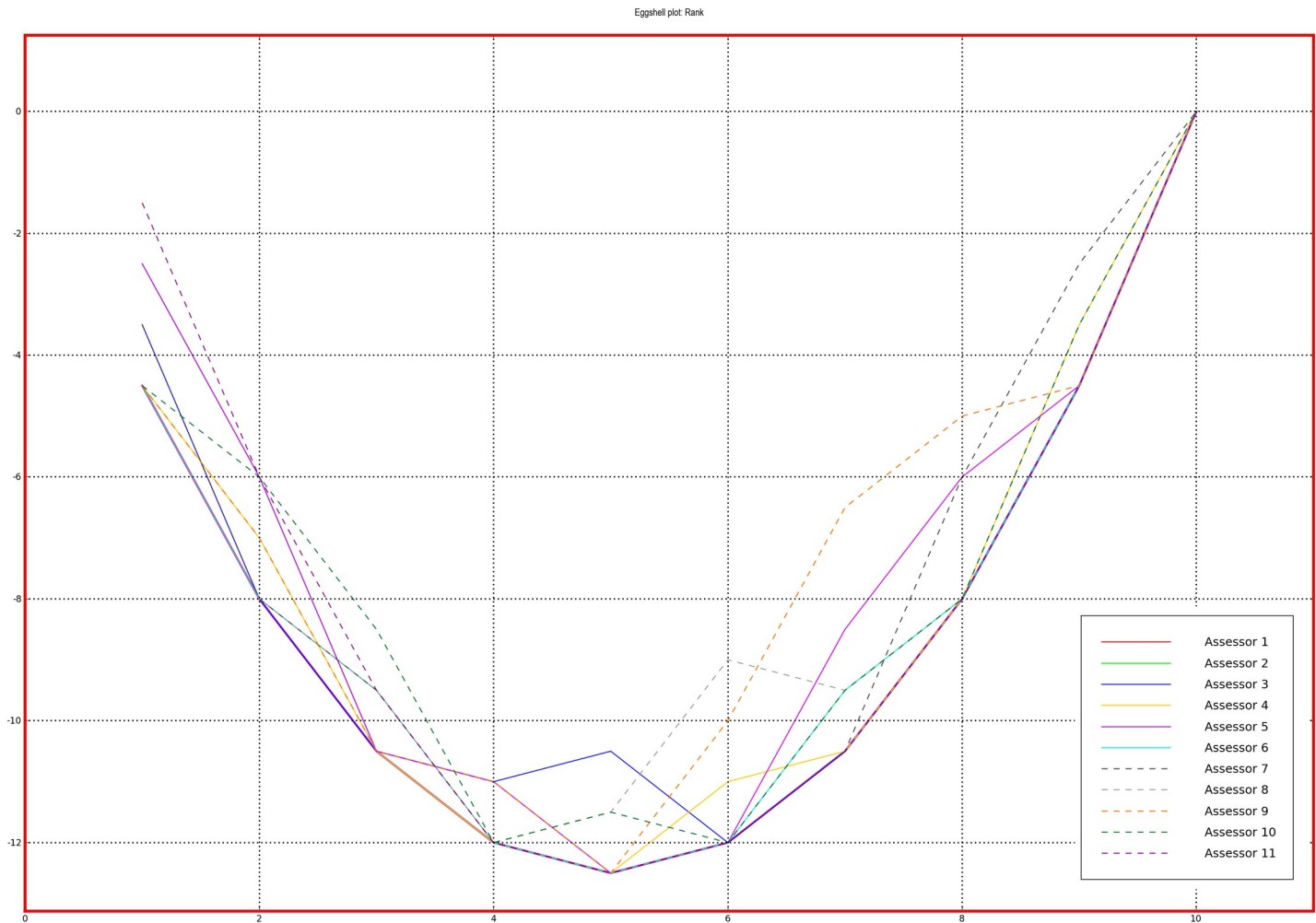

Eggshell plot: Rank

**Fig 4. Eggshell plot for each of the assessors–stage 2.**

consensus ranks vs. individual cumulative ranks, thus highlighting ranking differences among assessors. Each assessor is represented by its own line and the consensus line in the eggshell plot is located as a baseline below the other lines. The samples along the horizontal axis are sorted by intensity of the averages (in this case—the average ranks) in increasing order from left to right [39].

As can be observed, after the second training, the assessments of more panelists agree, the graph is smoother and more similar to the shape of an eggshell. It is worth noting that both in the first and the second stage of the training, the most problematic was the assessment of the initial samples, which were characterized by the highest dilution of n-butanol and thus the lowest odor intensity.

## 7. Conclusions

Training and monitoring the assessors is time consuming and costly. Undoubtedly, the simplification of procedures could be beneficial for panel leaders and for the panelists themselves [40]. On the other hand, when building a sensory team that is to perform odor nuisance tests, it should be borne in mind that there is a high probability that the assessors will have to

encounter many different odors, while the assessment itself may require the ability to describe and identify different odor notes. That is why, when the assessments are more complex, there is a need for more intensive training of the panelists [41]. Therefore, if appropriate resources are available when creating the sensory panel that will assess odor nuisance, various substances should be introduced together with adequate sensory analysis training. Sometimes it seems reasonable to consider not only the requirements of the EN 13725 standard, which mainly focuses on the individual recognition of detection threshold of n-butanol. As some authors point out [42], considering n-butanol as a single reference odor is debatable, as sensitivity to this substance does not have to correlate with sensitivity to other odors. As shown in this paper, the introduction of additional odors, and the development of other methods of training can contribute to greater consistency of sensory assessments.

Considering that in recent years, the attention of citizens towards air quality has increased significantly, and [32] that industrial odors will increasingly come under scrutiny, the need for properly trained assessors who can effectively conduct field inspections and provide unbiased information on odor nuisance is becoming ever more apparent.

## Supporting information

**S1 Appendix. This is the values used to build graphs.**
(XLSX)

## Author Contributions

**Conceptualization:** Paweł Turek.

**Data curation:** Paweł Turek.

**Formal analysis:** Paweł Turek.

**Funding acquisition:** Paweł Turek.

**Investigation:** Paweł Turek.

**Methodology:** Paweł Turek.

**Project administration:** Paweł Turek.

**Resources:** Paweł Turek.

**Software:** Paweł Turek.

**Supervision:** Paweł Turek.

**Validation:** Paweł Turek.

**Visualization:** Paweł Turek.

**Writing – original draft:** Paweł Turek.

**Writing – review & editing:** Paweł Turek.

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
