## [Decision Letter · Decision Letter 0]

19 Jul 2021

PONE-D-21-12936

Recruiting, Training and Managing a Sensory Panel in Odor Nuisance Testing

PLOS ONE

Dear Dr. Pawel,

Thank you for submitting your manuscript to PLOS ONE. After careful consideration, we feel that it has merit but does not fully meet PLOS ONE’s publication criteria as it currently stands. Therefore, we invite you to submit a revised version of the manuscript that addresses the points raised during the review process.

Please submit your revised manuscript by 10.08.2021. If you will need more time than this to complete your revisions, please reply to this message or contact the journal office at plosone@plos.org. Please include the following items when submitting your revised manuscript:

We look forward to receiving your revised manuscript.

Kind regards,

Prakash Kumar Sarangi

Academic Editor

PLOS ONE

Additional Editor Comments:

major revision

Journal Requirements:

Reviewers' comments:

Reviewer's Responses to Questions

**Comments to the Author**

1. Is the manuscript technically sound, and do the data support the conclusions?

Reviewer #1: Yes

Reviewer #2: No

2. Has the statistical analysis been performed appropriately and rigorously? 

Reviewer #1: Yes

Reviewer #2: Yes

3. Have the authors made all data underlying the findings in their manuscript fully available?

Reviewer #1: Yes

Reviewer #2: No

4. Is the manuscript presented in an intelligible fashion and written in standard English?

Reviewer #1: Yes

Reviewer #2: Yes

5. Review Comments to the Author

Reviewer #1: The paper represents a novel work for the society. Sometimes, the awareness is important than analysis for odor nuisance things. Authors should mention about future scope of work for this article.

Hence it is highly recommended for publication with minor revisions.

Reviewer #2: Reviewer comments

Overall this research objective is good and author tried to do some Odor Nuisance Testing, But there are some points to be considered before publishing this papers

1-Authors focused to Recruitment and Final selection (Qualification) approaches on panelist members based on duration of training periods that can do odor nuisance testing. But, there is no proper proof on scientific basis that can ensure for different panels for same food or non-product odor. So, author has to put some efforts/ evidences on this query

2-What are software used for creating these figure and how this odor testing procedure can be suitable for other products that contained many odor compound. This paper only mentioned few examples like n-butanol.

3-There are many old references, that is not suitable for current form of this paper, Go for new references that can support this objects

6. PLOS authors have the option to publish the peer review history of their article (what does this mean?). If published, this will include your full peer review and any attached files.

Reviewer #1: No

Reviewer #2: No

---

## [Author Response · Author response to Decision Letter 0]

16 Aug 2021

Response to the Reviewers

Reviewer 1

The paper shows a good suggestive work which is essential for Odor Nuisance Testing purpose. It is recommended for publication subjected to following minor revisions.

1. In the first paragraph of the introduction section, the authors are advised to mention more details about PN-EN 13725:2007 standard for more clarity.

As written in the first paragraph, this paper does not focus on detailed checks of sensory sensitivity and the principles of selecting candidates as described in the EN 13725 standard (since it provides clear guidelines within the said scope); instead, the emphasis is placed on the requirements included in the scientific standards of sensory analysis.

Thank you for bringing forward this issue. Following the reviewer’s comment, part 2 of the paper (Recruitment) has been modified and the more detailed requirements of the code of conduct for Assessors and Assessment Team Members, as set out in the clause 6.7.1 of the EN 13725 standard, are now presented.

2. Further, the authors are advised to mention correct section heading numbers, like 3 for Final selection (Qualification) in place of 2.

Section heading numbers have been corrected.

3. The quality of figures is not good. Hence, the authors are advised to show better resolutions in the paper.

Figures of appropriate quality have been provided in separate files. 

4. The authors should also cite a few more of the latest referred papers.

Another literature review has been conducted with the focus on the latest papers on the subject; consequently, some new content has been added to the revised manuscript. (Reference no 3, 7, 8, 13, 17, 32)

Reviewer 2

Reviewer comments

Overall, this research objective is good and author tried to do some Odor Nuisance Testing, but there are some points to be considered before publishing this paper.

1. Authors focused in the Recruitment and Final selection (Qualification) approaches on panelist members based on duration of training periods that can do odor nuisance testing. But, there is no proper proof on scientific basis that can ensure for different panels for same food or non-product odor. So, author has to put some efforts/ evidences on this query

As the reviewer pointed out, the aim of the article was to present the selection and the training procedure for the sensory analysis team, which was created for the purposes of odor nuisance testing. Based on the author’s own experience, resulting from many years of conducting sensory studies, the methods of the analysis, which are dedicated mainly to food products, have been adapted and, consequently, a more extensive training of panelists was proposed and performed. The method of training presented in the manuscript goes far beyond the requirements specified in the EN 13725 standard and includes many other odors that the panelists might encounter. It should be borne in mind that the requirements given in EN 13725, although very detailed, focus only on a single odor (n-butanol) as it was assumed by the authors of the standard that the sensitivity to the reference material would be an indicator of the sensitivity to other substances. Yet, in the light of research in the field of sensory analysis, the selection procedure should be much more extensive. The proposed solution presented in this study concerns the odor nuisance testing team. In the case of research on food products, researchers have at their disposal many more scientific studies focusing on specialized training, which can then be adapted to specific production profiles and subject standards (e.g., ISO 8586:2012 Sensory analysis — General guidelines for the selection, training and monitoring of selected assessors and expert sensory assessors). In the initial stage of the sensory team formation, samples of widely available odors are used to train the panelists to correctly identify the stimuli. The ability to scale them is checked at a later stage. Chemicals and mixtures of known compositions are used to ensure that different panelists receive samples of the same odor and intensity.

2. What is the software used for creating these figure and how this odor testing procedure can be suitable for other products that contained many odor compound. This paper only mentioned a few examples like n-butanol.

For the purposes of data presentation and to visualize the panel performance, free software PanelCheck (http://www.panelcheck.com/) was used. This information has been added to the manuscript. 

In response to the reviewer's question whether such software can be employed for the visualization of other products: it is primarily used to visualize data on panel performance. It also serves to assess whether the panelists’ assessment it consistent and uniform i.e., on its basis, it can be determined whether the sensory team members are able to differentiate the assessed samples. In such case, the number of compounds in the sample is not important for the statistical analysis itself, although it can be expected that with more difficult tasks the uniformity of the panelists’ assessments will be lower. 

According to the EN 13725 standard, n-butanol is the only substance used to assess sensory sensitivity and to qualify for the odor nuisance testing team. The authors of the standard assumed that the sensitivity to the reference material would be an indicator of the sensitivity to other substances. Therefore, it was also used in this study to see how the team's performance would change after an extended training course in which other substances were also used.

3. There are many old references, that is not suitable for current form of this paper, Go for new references that can support this objects

The literature review has been updated and some new content related to the topic of the study has been added to the manuscript. (Reference no 3, 7, 8, 13, 17, 32)

---

## [Editor Report · Decision Letter 1]

17 Sep 2021

Recruiting, Training and Managing a Sensory Panel in Odor Nuisance Testing

PONE-D-21-12936R1

Dear Dr. Turek

We’re pleased to inform you that your manuscript has been judged scientifically suitable for publication and will be formally accepted for publication once it meets all outstanding technical requirements.

Kind regards,

Prakash Kumar Sarangi

Academic Editor

PLOS ONE

Additional Editor Comments (optional):

accept
---

## [Editor Report · Acceptance letter]

8 Oct 2021

PONE-D-21-12936R1 

Recruiting, Training and Managing a Sensory Panel in Odor Nuisance Testing 

Dear Dr. Turek:

I'm pleased to inform you that your manuscript has been deemed suitable for publication in PLOS ONE. Congratulations! Your manuscript is now with our production department. 

Kind regards, 

on behalf of

Dr. Prakash Kumar Sarangi 

Academic Editor

PLOS ONE